# AgentPO: Enhancing Multi-Agent Collaboration via Reinforcement Learning

**Lin Sun**[1,2]*, **Chuang Liu**[2], **Can Zhang**[2], **Yubin Wu**[2], **Weijia Lu**[2], **Ning Wu**[2]

[1]MatrixRobotics, Shanghai, China
[2]UAES AI Lab, Shanghai, China

## Abstract

Multi-Agent Systems (MAS) offer a powerful paradigm for solving complex problems through distributed reasoning and collaboration. However, their effectiveness is often hindered by the challenge of optimizing interactions among agents. To address this, we introduce AgentPO, a novel framework that directly optimizes agent collaboration. AgentPO employs reinforcement learning to train a specialized Collaborator agent, which refines its interaction policy to enhance overall system performance within a fixed multi-agent topology. We evaluated AgentPO on multiple mathematical reasoning tasks, where it consistently outperformed strong baselines. With Llama-3.2-3B-Instruct as the actor model, AgentPO achieves accuracy improvements of +1.8% and +7.2% over strong baselines like Role Assignment and EvoAgent, respectively. When using the larger Llama-3.1-8B-Instruct model, these gains increase to +5.6% and +11.3%. Crucially, AgentPO achieves these results with remarkable efficiency: it requires only 500 training samples and operates at just 7.8% of EvoAgent's inference cost, highlighting its superior scalability and practicality. The code is available at: https://github.com/sunlin-ai/agentpo.

## 1 Introduction

Drawing inspiration from human collaborative intelligence, Multi-Agent Systems (MAS) offers a powerful paradigm for addressing challenges beyond the reach of individual agents (Shinn et al., 2023; Qian et al., 2025; Wang et al., 2025). By enabling agents to collaborate and take on specialized roles, MAS unlock new frontiers in artificial intelligence (Jimenez et al., 2024; Singh et al., 2025; Guo et al., 2024). Yet, realizing this potential faces a central challenge: enabling agents to cooperate effectively and robustly (Cemri et al., 2025). While significant progress has been made in developing individual agent capabilities, such as reasoning (Wei et al., 2022; Yao et al., 2023b; Hu & Clune, 2023), memory (Zhang et al., 2024b; Lewis et al., 2020), and tool use (Schick et al., 2023; Qu et al., 2025), integrating these components into a cohesive system remains a formidable task.

Current approaches to MAS design broadly fall into two categories. The first relies on manual orchestration of agent workflows, a labor-intensive process that requires deep domain expertise and meticulous prompt engineering (Shinn et al., 2023; Qian et al., 2025; Wang et al., 2025). However, because LLM-based agents are inherently sensitive to prompt variations, this approach introduces cascading instability: minor performance fluctuations in one agent can easily propagate and amplify instability across the entire system (Verma et al., 2024; Liu et al., 2024; Zhou et al., 2024a). The second category employs automated methods to search for optimal interaction topologies (Hu et al., 2025; Zhang et al., 2025; Zhuge et al., 2024). Yet, due to the combinatorial explosion of possible topologies, such search strategies quickly become infeasible at scale and often fail to discover truly effective collaborative structures (Gao et al., 2025).

To address these challenges, we propose a different viewpoint. Rather than manually designing complex agent topologies or searching the vast and fragile space of interaction structures, we contend that a more robust path lies in optimizing the collaborative dynamics within a well-designed topology. This reframes the core research question from *"What is the optimal agent topology?"* to

---

*Correspondence to Lin Sun <sunlin@matrixrobotics.ai>. This work was done at UAES AI Lab.

*"Given an effective topology, how can we train agents to cooperate more effectively to maximize system performance?"*

Building on this insight, we present AgentPO, an innovative framework that makes LLM agents better cooperate. Figure 1 illustrates AgentPO further, it operates by creating a partnership between two agent types: a lightweight Collaborator and a powerful Actor. The Actor, which can be a state-of-the-art model, focuses on task execution, while the Collaborator learns how to help it succeed. Crucially, we only train the small Collaborator. Using a simple success-or-fail reward signal, it learns to provide effective hints, critiques, or suggestions to its Actor partner. This approach boosts the team's overall efficiency and adaptability without the heavy computational cost of architectural search or fine-tuning powerful but ultra-large Actor models, offering a scalable and resource-efficient path to building cooperative MAS.

We evaluate AgentPO on multiple challenging mathematical reasoning benchmarks. Our experiments demonstrate that directly optimizing collaborative dynamics significantly enhances performance. For example, on the AMC23 benchmark, AgentPO achieves a +5.0% improvement in Pass@1 accuracy under the Hint-Actor topology and a +7.5% improvement under the Critic-Actor topology, substantially outperforming systems that rely on architectural search or standard Actor fine-tuning. Our main contributions are:

- We propose AgentPO, a novel framework that focuses on optimizing collaboration among agents through reinforcement learning.
- We present empirical results on complex reasoning tasks, showing that AgentPO achieves significantly enhanced performance, validating our hypothesis that optimizing collaborative dynamics is an effective strategy for improving system-level capabilities.
- We offer a practical and scalable methodology for enhancing multi-agent systems: by training small, specialized Collaborator within a well-designed topology, AgentPO achieves superior performance at a significantly lower cost.

## 2 METHODOLOGY

### 2.1 PROBLEM FORMULATION

We formalize the multi-agent collaborative problem-solving task as follows. Given a problem distribution $\mathcal{D}$, where each instance is a tuple $(q, y)$ comprising a problem description $q$ and its ground-truth solution $y$, the objective is to train a system of agents to collaboratively produce a solution $\hat{y}$ that maximizes the expected reward over this distribution:

$$\theta^* = \arg\max_{\theta} \mathbb{E}_{(q,y)\sim\mathcal{D}} \left[ R(\hat{y}, y) \right]. \tag{1}$$

where $R$ is the reward function.

### 2.2 THE AGENTPO FRAMEWORK

AgentPO's architecture is founded on a functional decoupling of collaboration and execution, implemented through two distinct roles: Collaborator Agent and Actor Agent. The Collaborator, governed by a learnable policy $\pi_\theta$ with parameters $\theta$, is the target of optimization. In contrast, the Actor operates with a fixed, high-performance policy $\pi_\phi$ using frozen parameters $\phi$. Our objective is to optimize $\theta$ to maximize the system's joint performance.

For any given problem $q$, the interaction proceeds sequentially. First, the Collaborator Agent generates an auxiliary signal $z$ conditioned on a problem-derived context:

$$z \sim \pi_\theta(\cdot \mid c_\theta(q)). \tag{2}$$

The composition of this context $c_\theta(q)$ is determined by the specific interaction topology (detailed in Sec.2.3). This signal $z$ is then used to form an enriched context for the Actor Agent, which in turn produces the final solution $\hat{y}$:

$$\hat{y} \sim \pi_\phi(\cdot \mid q, z). \tag{3}$$

Finally, the solution $\hat{y}$ is evaluated against the ground-truth $y$ to produce a binary reward:

$$R(\hat{y}, y) = \mathbb{I}(\hat{y} = y). \tag{4}$$

where $\mathbb{I}(\cdot)$ is the indicator function. This scalar reward provides the learning signal to update $\pi_\theta$ via our proposed objective (Sec. 2.4). This decoupling enables AgentPO to optimize for specialized collaborative skills within the interaction topology, enhancing system performance without altering the foundational capabilities of the Actor.

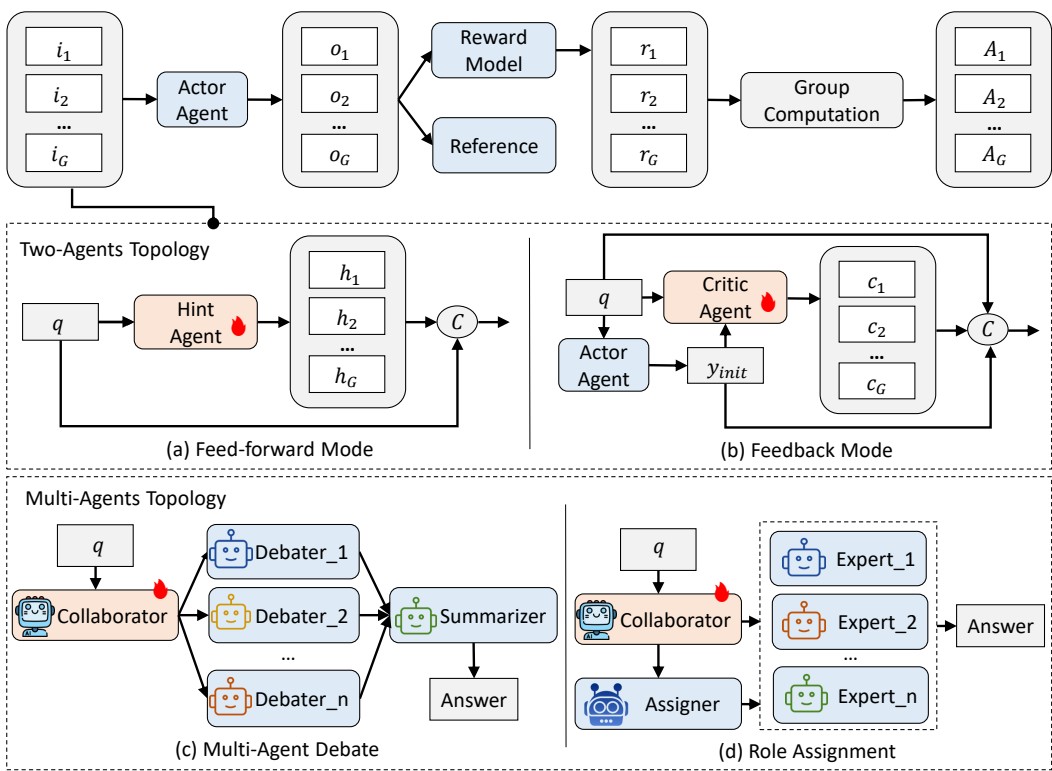

Figure 1: Our proposed AgentPO framework enhances multi-agent system performance through collaborative optimization. Given a problem, auxiliary signal is generated by the collaborator, which depends on the topology structure: the Hint Agent in the Feed-forward Mode, the Critic Agent in the Feedback Mode or the Collaborator in the Multi-Agents Topology. This auxiliary signal is then concatenated with the original problem to form the input for the Actor Agent. The Collaborator's parameters are subsequently optimized using the GRPO algorithm.

## 2.3 AGENT TOPOLOGIES

The interaction protocol between the Collaborator and Actor is defined by agent topology. A topology specifies two key components: (1) the input context $c_\theta$ for the Collaborator's policy $\pi_\theta$, and (2) the mechanism for integrating the Collaborator's output signal $z$ into the actor's context. We adopt and evaluate two representative and widely used topologies that embody two distinct collaboration paradigms: the feed-forward mode and the feedback mode, as is shown in Figure 1. Other topologies can be similarly defined and analyzed following this framework.

**AgentPO under Feed-forward Mode.** In this configuration, the Collaborator acts as a Hint Agent, providing proactive guidance. Conditioned solely on the problem $q$, it generates a hint $h \sim \pi_\theta(\cdot \mid q)$. This hint is then prepended to the problem query to form an augmented context for the Actor Agent, which then produces the final solution $y \sim \pi_\phi(\cdot \mid [q; h])$. The resulting reward trains the Hint Agent to generate maximally effective guidance. This topology embodies a feed-forward, proactive model of collaboration.

**AgentPO under Feedback Mode.** Here, the Collaborator assumes the role of a Critic Agent, facilitating an iterative refinement loop. The sequence begins with the Actor generating an initial draft solution, $y_{\text{init}} \sim \pi_\phi(\cdot \mid q)$. This draft, along with the problem $q$, forms the context for the Critic to produce a critique, $c \sim \pi_\theta(\cdot \mid [q; y_{\text{init}}])$. Finally, the Actor conditions on the complete history—the problem, its initial attempt, and the critique, to generate a refined solution, $y_{\text{ref}} \sim \pi_\phi(\cdot \mid [q; y_{\text{init}}; c])$. The reward, computed from $y_{\text{ref}}$, trains the Critic to provide feedback that most effectively steers the refinement process toward a correct solution. This topology models a reflective, feedback-driven form of collaboration.

## 2.4 MODULE OPTIMIZATION VIA GRPO

We optimize the Collaborator's parameters $\theta$ using Group Relative Policy Optimization (GRPO) (Shao et al., 2024), a policy gradient method designed for multi-output, solution-level reward settings. GRPO improves training stability by using the average reward within a group of outputs as a baseline, eliminating the need for a separate value function. For each input $q$, GRPO samples a group of responses $\{o_i\}_{i=1}^G$ from the current policy $\pi_{\theta_{\text{old}}}$ and computes advantages $\hat{A}_{i,t}$ based on the relative performance of each output within the group. The GRPO objective is defined as:

$$\mathcal{J}_{\text{GRPO}}(\theta) = \mathbb{E}_{q \sim \mathcal{D}, \{o_i\}_{i=1}^G \sim \pi_{\theta_{\text{old}}}(\cdot|q)}$$
$$\left[ \frac{1}{G} \sum_{i=1}^G \frac{1}{|o_i|} \sum_{t=1}^{|o_i|} \left( \min \left( r_{i,t}(\theta) \hat{A}_{i,t}, \text{clip}\left(r_{i,t}(\theta), 1-\varepsilon, 1+\varepsilon\right) \hat{A}_{i,t} \right) - \beta \mathbb{D}_{\text{KL}}(\pi_\theta \| \pi_{\text{ref}}) \right) \right].$$
$$(5)$$

where $r_{i,t}(\theta) = \frac{\pi_\theta(o_{i,t}|q,o_{i<t})}{\pi_{\theta_{\text{old}}}(o_{i,t}|q,o_{i<t})}$ is the probability ratio for token $o_{i,t}$, and $\hat{A}_{i,t}$ is the advantage estimated from intra-group reward comparisons, the KL divergence term $\mathbb{D}_{\text{KL}}(\pi_\theta \| \pi_{\text{ref}})$ regularizes policy updates relative to a reference policy $\pi_{\text{ref}}$. By maximizing $\mathcal{J}_{\text{GRPO}}(\theta)$, the Collaborator learns to generate signals, such as hints or critiques, that effectively guide the Actor toward higher-reward solutions, thereby acquiring a robust and generalizable collaborative policy.

# 3 EXPERIMENTS

## 3.1 SETUP

**Datasets.** For training, we use the MATH dataset (Hendrycks et al., 2021), focusing on problems from difficulty levels 3 to 5, following the same setup as prior work (Liu et al., 2025). For evaluation, we benchmark our method across five mathematical reasoning datasets: (1) **AIME24**: A collection of 30 high-school olympiad-level problems from the 2024 American Invitational Mathematics Examination (Li et al., 2024). (2) **AMC**: A set of 83 intermediate-difficulty problems from the American Mathematics Competitions, primarily in multiple-choice format (Li et al., 2024). (3) **MATH500**: A randomly sampled subset of 500 problems from the MATH dataset, spanning algebra, geometry, and number theory (Hendrycks et al., 2021). (4) **MinervaMath**: A benchmark of 272 multi-step reasoning problems (Lewkowycz et al., 2022). (5) **OlympiadBench**: A challenging suite of 675 high-difficulty mathematics problems (He et al., 2024).

**Models and Baselines.** Our experiments employ several models from the Llama and Qwen families, including `Llama-3.2-3B-Instruct`, `Llama-3.1-8B-Instruct` (Dubey et al., 2024), `Qwen2.5-3B-Instruct`, `Qwen2.5-7B-Instruct` (Yang et al., 2024a), `Qwen2.5-Math-7B` (Yang et al., 2024b) and `Qwen-Plus` (Yang et al., 2024a). Among these, `Qwen2.5-3B-Instruct` and `Qwen2.5-7B-Instruct` are employed as the Collaborator, while the remaining LLMs serve as the Actor. We compare AgentPO against several strong baselines: (1) **CoT**: Zero-shot Chain-of-Thought prompting (Wei et al., 2022). (2) **Self-Consistency**: CoT with self-consistency, which selects the most frequent answer from multiple reasoning paths (Wang et al., 2023). (3) **Self-Refine**: An iterative method where agents critique and refine their own outputs (Madaan et al., 2023; Shinn et al., 2023). (4) **Multi-Agent Debate**: A framework in which agents debate their answers and aggregate peer feedback (Du et al., 2024). (5) **Step-back Abstraction**: a prompting method that encourages agents to first reason about the core principles underlying

a task, enhancing overall reasoning effectiveness (Zheng et al., 2024). (6) **Quality-Diversity**: an approach that generates and ensembles diverse solution candidates to better explore the solution space (Lu et al., 2025). (7) **Role Assignment**: a strategy that assigns distinct roles to foundation models to promote specialized thinking and improve response quality (Xu et al., 2023). (8) **EvoAgent**: a self-evolving agent that autonomously performs long-horizon tasks through self-planning, self-control, and self-reflection (Feng et al., 2025).

**Evaluation Metrics.**  Following established practice (Liu et al., 2025; Zeng et al., 2025), our primary metric is **Pass@1** (Chen et al., 2021). Pass@k measures whether at least one of $k$ independently generated solutions is correct. We focus on the more strict Pass@1 setting, which evaluates the accuracy of a single generated response and serves as a robust indicator of model reliability.

**Implementation Details.**  We conduct reinforcement learning training using the `verl` framework (Sheng et al., 2025). We set the clipping threshold to $\varepsilon = 0.2$, and during training, we sample 16 rollouts per prompt at a temperature of 1.0, with a maximum response length of 2048 tokens. The global batch size is 16, with a per-GPU mini-batch size of 4 and a learning rate of $1 \times 10^{-6}$. For inference, we use the vLLM library (Kwon et al., 2023), setting temperature to 0.0 and top-p to 1.0. To ensure rigorous evaluation on mathematical problems, we incorporate verification functions from Math-Verify. All experiments are conducted on a cluster of 1 compute node, equipped with 4 NVIDIA A40 40GB GPUs. We re-implemented multiple baselines for comparison with AgentPO, details are in Appendix C.

## 3.2 MAIN RESULTS

The primary experimental results are presented in Table 1, obtained using the Hint-Actor topology with Qwen2.5-3B-Instruct as the Hint model. Due to the support for long contexts (up to 128K tokens), Llama-family models are used as the Actor model in our comparative analysis.

Table 1: **Performance Comparison of Various Baselines on Multiple Benchmarks.** AgentPO is evaluated using the Hint-Actor topology, with Qwen2.5-3B-Instruct as the hint model. **Avg.** indicates mean accuracy across all test datasets. Top results are in **bold**, and runner-up results are underlined. Performance improvements ($\Delta$) are relative to each baseline method.

| Algorithm | AIME24 | Math500 | OlympiadBench | Minerva | AMC23 | Avg. | $\Delta$ |
|---|---|---|---|---|---|---|---|
| **Actor Model: Llama-3.2-3B-Instruct** | | | | | | | |
| CoT | 6.7 | 48.4 | 16.4 | 21.0 | 15.0 | 21.5 | +3.0 |
| Self-Consistency | 3.3 | 49.6 | 15.9 | 16.5 | **27.5** | 22.6 | +1.9 |
| Self-Refine | **10.0** | 33.6 | 12.7 | 15.1 | 12.5 | 16.8 | +7.7 |
| Multi-Agent Debate | 3.3 | 47.2 | 15.7 | 20.2 | 25.0 | 22.3 | +2.2 |
| Step-back Abstraction | 10.0 | 43.0 | 13.9 | 14.0 | 20.0 | 20.2 | +4.3 |
| Quality-Diversity | 3.3 | 40.0 | 14.2 | 15.1 | 17.5 | 18.0 | +6.5 |
| Role Assignment | 10.0 | 49.4 | **16.6** | 17.6 | 20.0 | 22.7 | +1.8 |
| EvoAgent | 10.0 | 33.4 | 12.6 | 13.2 | 17.5 | 17.3 | +7.2 |
| AgentPO (ours) | 6.7 | **52.2** | 16.1 | **22.4** | 25.0 | **24.5** | |
| **Actor Model: Llama-3.1-8B-Instruct** | | | | | | | |
| CoT | 3.3 | 43.0 | 15.0 | 18.4 | 17.5 | 19.4 | +12.1 |
| Self-Consistency | 3.3 | 50.6 | 17.2 | 19.5 | 25.0 | 23.1 | +8.4 |
| Self-Refine | 0.0 | 43.2 | 15.0 | 18.0 | 30.0 | 21.2 | +10.3 |
| Multi-Agent Debate | 6.7 | 50.2 | 15.6 | 24.3 | 25.0 | 24.4 | +7.1 |
| Step-back Abstraction | 3.3 | 44.2 | 14.4 | 18.8 | 20.0 | 20.1 | +11.4 |
| Quality-Diversity | 3.3 | 45.2 | 16.3 | 19.5 | 17.5 | 20.4 | +11.1 |
| Role Assignment | 6.7 | 47.4 | 15.9 | 24.3 | **35.0** | 25.9 | +5.6 |
| EvoAgent | 3.3 | 36.0 | 12.7 | 16.5 | 32.5 | 20.2 | +11.3 |
| AgentPO (ours) | **16.7** | **60.6** | **28.9** | **26.1** | 25.0 | **31.5** | |

**Comparison with Baseline Methods.**  Our proposed method, AgentPO, consistently outperforms all baseline algorithms. When employing the Llama-3.2-3B-Instruct as Actor model, AgentPO

achieves an average accuracy of 24.5%, representing a 1.8% improvement over the strong baseline Role Assignment, and demonstrates a significant improvement of up to 7.7% over methods like Self-Refine. This performance margin becomes even more pronounced with the more powerful Llama-3.1-8B-Instruct model. In this setting, AgentPO attains an average accuracy of 31.5%, widening the gap with the top-performing baseline, Role Assignment to 5.6%. Significantly, our method surpasses conventional Chain-of-Thought prompting by a considerable margin of 12.1%. These results robustly validate the superior efficacy of the AgentPO framework in enhancing the reasoning capabilities.

**Scaling with Actor Model Capability.** A critical insight from our experiments is that the benefits of AgentPO scale effectively with the capability of the Actor model. By upgrading the Actor model from Llama-3.2-3B-Instruct to Llama-3.1-8B-Instruct, the average performance of AgentPO increased from 24.5% to 31.5%, representing a substantial improvement of 7.0%. This scalability is particularly pronounced on challenging benchmarks; for example, accuracy on AIME24 more than doubled from 6.7% to 16.7%, and on OlympiadBench, it increased from 16.1% to 28.9%. These findings suggest that a more advanced Actor model provides a stronger foundation for system-level optimization, enabling the discovery and utilization of effective collaborative strategies, and thereby unlocking the latent reasoning capabilities of large language models.

## 3.3 ANALYSIS

**Impact of Collaborator Model Selection.** To evaluate the impact of the Collaborator model, we use the Hint-Actor topology, varying the Hint model while keeping the Actor model fixed. As shown in Figure 2, AgentPO demonstrates strong robustness, consistently improving performance across different Hint models. Interestingly, we find that the smaller Qwen2.5-3B-Instruct achieves the highest accuracy when paired with the Llama-3.1-8B-Instruct Actor, outperforming the larger Qwen2.5-7B-Instruct. Moreover, Qwen-series models generally yield better results, which we attribute to complementary strengths. Specifically, Qwen generates hints with reasoning patterns distinct from those of Llama, providing diverse insights that help the Actor overcome inherent biases. This underscores a key principle in multi-agent systems: effective collaboration depends not solely on individual model capacity, but on the synergy of complementary capabilities.

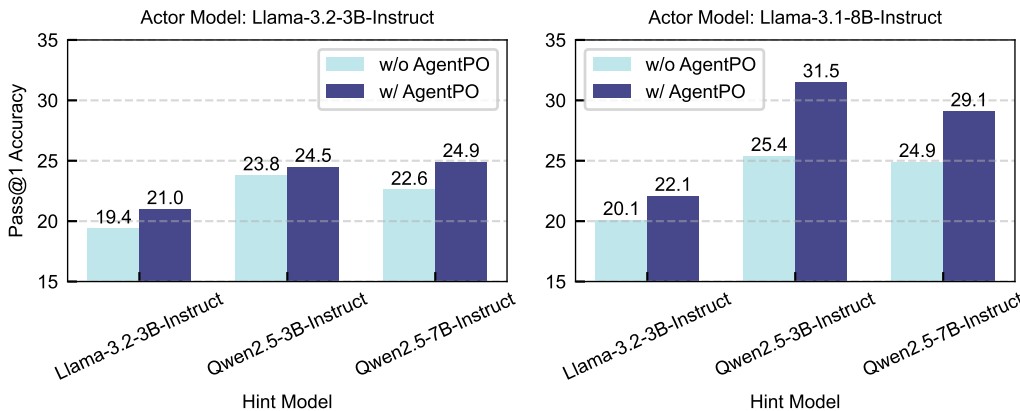

Figure 2: **Impact of Collaborator Model Choice.** AgentPO is evaluated using Hint-Actor topology with varying Hint models. AgentPO shows strong robustness, consistently enhancing performance across various Hint models, with Qwen-series models achieving the best results.

**Collaborative vs. Actor Optimization.** A key question is whether collaborative optimization can match or exceed direct Actor fine-tuning. To investigate this, we compare AgentPO against various Actor fine-tuning baselines, using the widely adopted Qwen2.5-Math-7B as the Actor model in all experiments. In AgentPO, we employ the lightweight Hint-Actor topology, using Qwen2.5-3B-Instruct as the Hint model. Crucially, only the 3B Hint model is finetuned in our method, while the baselines finetune the full 7B Actor model. As shown in Table 2, AgentPO achieves 49.4%

Table 2: **Comparsion between Collaborative and Actor Optimization.** AgentPO is evaluated using the Hint-Actor topology with Qwen2.5-3B-Instruct as Hint model and Qwen2.5-Math-7B as Actor model. Results show that collaborative optimization outperforms direct Actor fine-tuning.

| Algorithm | AIME24 | Math500 | OlympiadBench | Minerva | AMC23 | Avg. |
|---|---|---|---|---|---|---|
| Qwen2.5-Math-7B | 0.2 | 69.0 | 34.7 | 21.3 | 45.8 | 38.2 |
| **Actor Optimization** | | | | | | |
| SimpleRL-Zero-7B | 26.7 | 78.2 | 40.3 | 27.6 | 60.2 | 46.6 |
| Prime-Zero-7B | 16.7 | 83.8 | 40.9 | 36.0 | 62.7 | 48.0 |
| OpenReasoner-Zero-7B | 13.3 | 79.2 | 44.0 | 31.6 | 47.0 | 43.0 |
| **Collaborative Optimization** | | | | | | |
| AgentPO (ours) | 13.3 | 83.8 | 39.1 | 43.4 | 67.5 | **49.4** |

average accuracy, surpassing all baselines at lower training cost. This shows that (i) collaborative optimization can match or even exceed direct Actor optimization, and (ii) finetuning a lightweight Collaborator is more efficient than refining the expert itself.

**Data Efficiency of Collaborative Optimization.** We evaluate AgentPO's data efficiency under the Hint-Actor topology, using Qwen2.5-3B-Instruct as the hint model and Qwen2.5-Math-7B as the actor model, with training sets from 100 to 1,000 samples. As shown in Figure 3, AgentPO achieves 45.5% average accuracy with only 100 samples and reaches 49.4% with 500, demonstrating rapid learning of effective collaboration. In contrast, direct Actor optimization typically requires over 10,000 samples for comparable performance. A slight performance dip is observed at 700 and 1,000 samples. We hypothesize this stems from mild policy overfitting: as training progresses, the hint agent begins to over-adapt to idiosyncrasies in the limited training data, reducing its generalization capability. We posit that AgentPO's ef-

Table 3: **Performance of Hybrid Multi-Agent Systems.** AgentPO is trained using the Hint-Actor topology, with only the local Qwen2.5-7B-Instruct model updated as the hint generator, while an API-based Qwen-Plus serves as the fixed actor model.

| Algorithm | Base | w/ AgentPO | Δ |
|---|---|---|---|
| AIME24 | 33.3 | **36.7** | +3.4 |
| Math500 | 84.0 | **85.4** | +1.4 |
| OlympiadBench | 49.6 | 49.6 | 0.0 |
| Minerva | 46.3 | **47.4** | +1.1 |
| AMC23 | 70.0 | **75.0** | +5.0 |
| Avg. | 56.6 | **58.8** | +2.2 |

ficiency stems from learning a meta-skill, how to guide a capable expert, rather than domain knowledge from scratch. Consequently, AgentPO offers a practical and scalable path to improved LLM reasoning, especially in data-scarce domains.

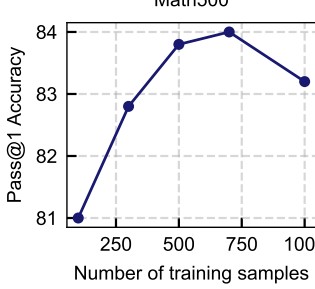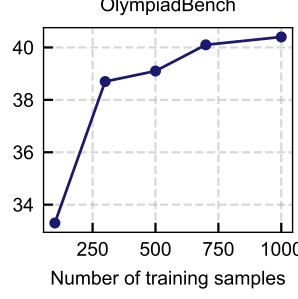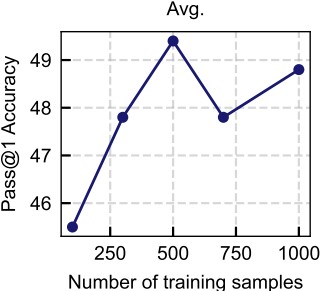

Figure 3: **Impact of Training Set Size on Performance.** AgentPO is evaluated using Hint-Actor topology with Qwen2.5-3B-Instruct as the Hint model and Qwen2.5-Math-7B as the Actor model. Results show that average performance peaks with only 500 training samples, indicating rapid learning of effective collaborative strategies.

**Local Collaborator Enhances API-based Powerful Actor.** We present a compelling application of AgentPO: building hybrid multi-agent systems that combine powerful, black-box API-based models with lightweight, locally trained agents. While these large models offer state-of-the-art capabilities, they cannot be fine-tuned directly. AgentPO overcomes this limitation by training a small, open-source model as a specialized Collaborator that strategically guides the API-based Actor model to improve its performance. Results in Table 5 demonstrate the effectiveness of this approach. When Qwen-Plus is used as the API-based Actor, its standalone accuracy is 56.6%. With AgentPO and a locally trained Qwen2.5-7B-Instruct Hint model, performance increases to 58.8%. This shows that a compact, cost-efficient local model can effectively co-pilot a large, inflexible model through targeted guidance, delivering measurable gains. The hybrid paradigm enables more powerful, customizable, and economically efficient AI systems, enhancing top-tier models without requiring access to their internal parameters.

Table 4: **Performance under Different Collaboration Topologies.** Two topologies are evaluated using Qwen2.5-7B-Instruct as the Collaborator model and Qwen2.5-Plus as the fixed Actor model. Performance improvements ($\Delta$) are relative to Qwen-Plus.

| Algorithm | AIME24 | Math500 | OlympiadBench | Minerva | AMC23 | Avg. | $\Delta$ |
|---|---|---|---|---|---|---|---|
| Qwen-Plus | 33.3 | 84 | 49.6 | 46.3 | 70 | 56.6 | - |
| **Hint-Actor** | | | | | | | |
| w/o AgentPO | 26.7 | 85.4 | 48.6 | 44.9 | 70 | 55.1 | -1.5 |
| w/ AgentPO | 36.7 | 85.4 | 49.6 | 47.4 | 75 | 58.8 | +2.2 |
| **Critic-Actor** | | | | | | | |
| w/o AgentPO | 36.7 | 86 | 51.4 | 45.2 | 70 | 57.9 | +1.3 |
| w/ AgentPO | 36.7 | 84.6 | 52.3 | 43.4 | 77.5 | 58.9 | +2.3 |

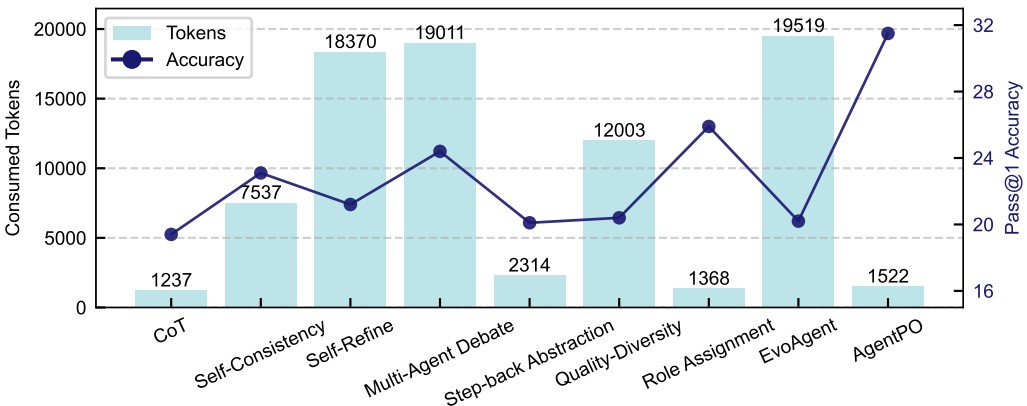

Figure 4: **Accuracy vs. Inference Cost.** AgentPO achieves the highest Pass@1 accuracy while using significantly fewer tokens, with consumed tokens on the primary axis and Pass@1 accuracy on the secondary axis, demonstrating a superior efficiency-accuracy trade-off compared to baseline methods.

**Comparison of Agent Collaboration Topologies** To evaluate the impact of collaboration structure, we compare two topologies: Hint-Actor and Critic-Actor. In both, Qwen2.5-7B-Instruct serves as the trainable Collaborator (Hint or Critic), while Qwen-Plus remains a frozen Actor. Results in Table 4 reveal how interaction topology and AgentPO optimization jointly shape performance. In the Hint-Actor setup, the unoptimized version (w/o AgentPO) degrades performance to 55.1%, below the standalone Qwen-Plus baseline, indicating that untrained hints can mislead rather than help. With AgentPO optimization, performance rises sharply to 58.8%, transforming ineffective assistance into a significant gain, highlighting the necessity of system-level alignment. In contrast,

the Critic-Actor topology performs well even without optimization, suggesting that critique-based reasoning is inherently effective. AgentPO further improves performance to 58.9%, the highest result in our experiments. These results confirm AgentPO's flexibility across collaboration topology and underscore a key principle: effective collaboration requires not just structure, but deliberate optimization.

**Extending to Complex Multi-Agent Topologies** To assess AgentPO's applicability beyond pairwise interactions, we evaluate it in two structured multi-agent systems: Multi-Agent Debate and Dynamic Role Assignment. In both settings, a lightweight Qwen2.5-3B-Instruct model is trained as the Collaborator, while Llama-3.2-3B-Instruct serves as the frozen Actor; the underlying multi-agent protocols remain unchanged. Results in Table 5 show that AgentPO consistently enhances performance across benchmarks. These results demonstrate that even in complex, fixed multi-agent topologies, optimizing a minimal collaborative signal is sufficient to unlock significant gains.

Table 5: **Performance of Multi-Agent Systems with More Agents.** AgentPO uses Hint-Actor topology: Qwen2.5-3B-Instruct as hint generator (trained), Llama-3.2-3B-Instruct as fixed actor. All results report mean ± standard deviation over five random seeds.

| Algorithm | AIME24 | Math500 | OlympiadBench |
|---|---|---|---|
| Debate | 8.9 ± 3.8 | 46.3 ± 1.0 | 15.3 ± 1.2 |
| w/ AgentPO | **12.3 ± 2.6** | **49.6 ± 0.8** | **21.1 ± 1.4** |
| Δ | +3.4 | +3.3 | +5.8 |
| Role Assignment | 7.3 ± 2.9 | 48.3 ± 0.8 | 15.7 ± 0.6 |
| w/ AgentPO | **10.2 ± 1.7** | **52.2 ± 0.6** | **20.9 ± 0.8** |
| Δ | +2.9 | +3.9 | +5.2 |

**Inference Efficiency and Practicality.** Inference cost measured in token consumption is a key factor for real-world deployment. We evaluate the trade-off between accuracy and token usage across methods. Results in Figure 4 show that AgentPO achieves top-tier performance at remarkably low cost. With Llama-3.1-8B-Instruct, AgentPO reaches 31.5% accuracy with just 1522 tokens on average, far below Self-Consistency and Multi-Agent Debate, which use 5–12× more tokens yet achieve lower accuracy. AgentPO is as efficient as lightweight methods like CoT and Role Assignment, but significantly outperforms them. AgentPO's efficiency stems from its design: collaboration is offloaded to a small, optimized Collaborator, enabling the powerful Actor to generate the final answer in a single, guided pass. By avoiding iterative or sampling-heavy processes, AgentPO breaks the traditional accuracy–cost trade-off, offering a scalable, cost effective solution for advanced reasoning in practice.

## 4 RELATED WORK

**Multi-Agent Systems.** The design of LLM-based Multi-Agent Systems typically unfolds at two levels: enhancing individual agent capabilities and orchestrating their interactions. At the agent level, key advances include Chain-of-Thought and tree-based reasoning for planning (Wei et al., 2022; Yao et al., 2023b;a), self-reflection for refinement (Madaan et al., 2023; Shinn et al., 2023), tool use (Qu et al., 2025; Schick et al., 2023; Nakano et al., 2021), and retrieval-augmented knowledge access (Lewis et al., 2020; Zhang et al., 2024b). At the system level, coordination strategies range from parallel exploration with aggregation (e.g., self-consistency (Madaan et al., 2023)) to sequential refinement and multi-agent debate, which enhances reasoning through argumentation (Chen et al., 2024; Wang et al., 2024; Zhang et al., 2024a). While these efforts establish foundational components and architectures, our work focuses on enabling effective cooperation within such systems.

**Automated Design and Optimization of MAS.** A parallel line of research focuses on automating the design of these complex systems. Optimization of multi-agent system has been explored across various dimensions, including prompts (Khattab et al., 2024), tools (Zhou et al., 2024b), and reasoning strategies (Shang et al., 2025). The most ambitious approaches focus on discovering the optimal workflow or topology. For instance, ADAS and AFlow leverage advanced search algorithms and LLM-based controllers to navigate the design space (Hu et al., 2025; Zhang et al., 2025), while GPTSwarm uses policy gradients to optimize the connections between agentic nodes (Zhuge et al., 2024). In contrast to these approaches, which primarily focus on searching the vast architectural space, AgentPO introduces a complementary paradigm: collaborative optimization within a fixed

topology. Instead of discovering what the workflow should be, we directly train agent modules on how to cooperate effectively, using system-level rewards to foster emergent teamwork.

## 5 CONCLUSION

We present AgentPO, a framework that shifts multi-agent system design from manual or costly architectural search to collaborative optimization. By fixing the system topology and training a lightweight Collaborator via reinforcement learning, AgentPO optimizes inter-agent cooperation end-to-end. Experiments on mathematical reasoning show consistent gains over strong baselines, demonstrating that optimizing interaction dynamics within a simple, fixed structure is more effective and scalable than exploring complex architectures. AgentPO enables efficient enhancement of powerful models through small, task-specific Collaborators. Future directions include integrating high-level architecture search and extending to dynamic, multi-agent systems with adaptive roles. AgentPO advances the path toward robust, adaptive, and truly collaborative AI for real-world problem-solving.

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

# A PROMPT TEMPLATE

We provide all prompt templates used for the collaborator and actor in both the Hint-Actor and Critic-Actor topologies. We use <> to enclose texts that have been skipped for presentation purposes.

## A.1 PROMPT FOR HINT-ACTOR TOPOLOGY

We use the following prompts for the Hint-Actor topology.

---

**Hint Agent Prompt**

\<Instruction\>
Rewrite the question below to make it easier to understand.

\<Problem\>
{{problem}}

---

**Actor Agent Prompt**

\<Instruction\>
Please reason step by step, and put your final answer within boxed{}.

\<Problem\>
{{problem (Hint: hint)}}

---

## A.2 PROMPT FOR CRITIC-ACTOR TOPOLOGY

We use the following prompts for the Critic-Actor topology.

---

**Critic Agent Prompt**

\<Instruction\>
Given a question and its current solution, analyze the solution and provide concise, specific feedback identifying any errors, logical gaps, or missing justifications. Do not rewrite the solution, only highlight issues.

\<Problem\>
{{problem}}

\<Current solution\>
{{current solution}}

---

**Actor Agent Prompt**

\<Instruction\>
You have an opportunity to improve your solution. Please review Current Solution and Comment carefully. Correct errors and fill justification gaps if any.

\<Problem\>
{{problem}}

\<Current solution\>
{{current solution}}

\<Comment\>
{{comment}}

---

## B AGENT ROLE

Table 6: Definitions of Agent Roles in Multi-Agent Systems.

| Agent Role | Function Description |
|---|---|
| Hint | Provides proactive suggestion or plan to guide the Actor. |
| Critic | Evaluates the Actor's output and offers feedback. |
| Actor | The primary reasoning agent that produces the final answer. |
| Collaborator | A trainable auxiliary agent (e.g., Hint or Critic) . |
| Debater | Participates in multi-agent debate by generating arguments. |
| Expert | A specialized agent assigned to a specific subtask. |

## C    BASELINES

In this paper, we implement eight agent baselines for comparison with AgentPO (Section 3.2). In CoT, The actor is prompted to think step by step before answering the question. In Self-Consistency, We sample N=3 responses and aggregate them using a final decision agent. In Self-Refine, The model performs up to three refinement iterations, with early stopping if the critic determines the answer is correct. In Multi-Agent Debate, Agents are assigned distinct roles, such as Reading Comprehension Specialist, Logical Reasoning Strategist, and Multidisciplinary Knowledge Integrator, and engage in a two-round debate. In Step-back Abstraction, A principle agent first identifies the underlying concepts of the task, followed by a CoT agent that uses this knowledge to answer the question. In Quality-Diversity, We run three iterations to generate diverse solutions based on previously proposed answers. In Role Assignment, A routing agent selects a role from a predefined set, and the corresponding expert agent answers the question in that role. In EvoAgent, Over two evolution rounds, one new agent is generated from each parent via crossover and mutation. Finally, all solutions are aggregated by a CoT agent. These baselines represent diverse strategies for multi-agent reasoning and collaboration, providing a comprehensive comparison with AgentPO.

## D    ADDITIONAL EXPERIMENTS

In this section, we provide further details on the experiments described in Section 3.3, along with a comprehensive analysis and comparison of the AgentPO algorithm across different data tasks.

Table 7: Analysis of training data size impact on model performance under the hint-actor topology. The collaborator model is Qwen2.5-3B-Instruct and the actor agent is Qwen2.5-Math-7B-Base. Reported values are pass@1 accuracy achieved with different amounts of training data.

| Training Set Number | AIME24 | Math500 | OlympiadBench | Minerva | AMC23 | Avg. |
|---|---|---|---|---|---|---|
| 100 | 13.3 | 81.0 | 33.3 | 44.9 | 55.0 | 45.5 |
| 300 | 10.0 | 82.8 | 38.7 | 42.3 | 65.0 | 47.8 |
| 500 | 13.3 | 83.8 | 39.1 | 43.4 | 67.5 | 49.4 |
| 700 | 10.0 | 84.0 | 40.1 | 42.3 | 62.5 | 47.8 |
| 1000 | 16.7 | 83.2 | 40.4 | 41.2 | 62.5 | 48.8 |

Table 8: Analysis of collaborator model impact on performance under the hint-actor topology. The actor agent uses Llama-3.2-3B-Instruct, while the collaborator model varies. Each model is evaluated with and without AgentPO: "w/o AgentPO" denotes the native multi-agent system without reinforcement learning training, and "w/ AgentPO" denotes the system after RL-based optimization. Reported values are pass@1 accuracy on the target tasks.

| Control | Actor | AIME24 | Math500 | OlympiadBench | Minerva | AMC23 | Avg. |
|---|---|---|---|---|---|---|---|
| Llama-3.2-3B | w/o AgentPO | 6.7 | 45.6 | 15.6 | 14.3 | 15.0 | 19.4 |
| | w/ AgentPO | 10.0 | 46.8 | 14.4 | 16.5 | 17.5 | 21.0 |
| Qwen2.5-3B | w/o AgentPO | 6.7 | 48.6 | 22.5 | 21.0 | 20.0 | 23.8 |
| | w/ AgentPO | 6.7 | 52.2 | 16.1 | 22.4 | 25.0 | 24.5 |
| Qwen2.5-7B | w/o AgentPO | 3.3 | 50.4 | 20.7 | 18.8 | 20.0 | 22.6 |
| | w/ AgentPO | 10.0 | 51.2 | 17.5 | 18.4 | 27.5 | 24.9 |

Table 9: Analysis of collaborator model impact on performance under the hint-actor topology. The actor agent uses Llama-3.1-8B-Instruct, while the collaborator model varies.

| Collaborator | Algorithm | AIME24 | Math500 | OlympiadBench | Minerva | AMC23 | Avg. |
|---|---|---|---|---|---|---|---|
| Llama-3.2-3B | w/o AgentPO | 3.3 | 46.2 | 15.9 | 15.1 | 20.0 | 20.1 |
| | w/ AgentPO | 10.0 | 48.4 | 15.3 | 19.1 | 17.5 | 22.1 |
| Qwen2.5-3B | w/o AgentPO | 16.7 | 48.0 | 21.2 | 21.0 | 20.0 | 25.4 |
| | w/ AgentPO | 16.7 | 60.6 | 28.9 | 26.1 | 25.0 | 31.5 |
| Qwen2.5-7B | w/o AgentPO | 10.0 | 46.4 | 20.1 | 22.8 | 25.0 | 24.9 |
| | w/ AgentPO | 13.3 | 60.8 | 29.5 | 24.6 | 17.5 | 29.1 |

Table 10: Token consumption across multiple mathematical tasks under the hint-actor topology. The hint agent uses Qwen2.5-3B-Instruct and the actor agent uses Llama-3.2-3B-Instruct. Token counts include all input and output tokens consumed during inference. All baseline methods are evaluated under the same setup for fair comparison.

| Algorithm | AIME24 | Math500 | OlympiadBench | Minerva | AMC23 | Avg. |
|---|---|---|---|---|---|---|
| CoT | 833 | 624 | 927 | 750 | 907 | 808 |
| Self-Consistency | 7038 | 4282 | 5942 | 4792 | 6394 | 5690 |
| Self-Refine | 14484 | 10711 | 18162 | 12945 | 13854 | 14031 |
| Multi-Agent Debate | 22448 | 12837 | 18172 | 13558 | 19564 | 17316 |
| Step-back Abstraction | 2474 | 1863 | 2354 | 1861 | 2600 | 2230 |
| Quality-Diversity | 13318 | 6642 | 13055 | 8097 | 11496 | 10522 |
| Role Assignment | 1250 | 1108 | 1313 | 996 | 1067 | 1147 |
| EvoAgent | 19130 | 9501 | 18215 | 10159 | 15069 | 14415 |
| AgentPO | 1166 | 747 | 1031 | 907 | 988 | 968 |

Table 11: Token consumption across multiple mathematical tasks under the hint-actor topology. The hint agent uses Qwen2.5-3B-Instruct and the actor agent uses Llama-3.1-8B-Instruct.

| Algorithm | AIME24 | Math500 | OlympiadBench | Minerva | AMC23 | Avg. |
|---|---|---|---|---|---|---|
| CoT | 1743 | 927 | 1454 | 916 | 1148 | 1237 |
| Self-Consistency | 9117 | 6457 | 7601 | 5342 | 9170 | 7537 |
| Self-Refine | 20614 | 15242 | 18410 | 16006 | 21578 | 18370 |
| Multi-Agent Debate | 24476 | 14683 | 18409 | 15702 | 21784 | 19011 |
| Step-back Abstraction | 2458 | 2334 | 2350 | 2216 | 2213 | 2314 |
| Quality-Diversity | 15065 | 9944 | 13439 | 9909 | 11657 | 12003 |
| Role Assignment | 1547 | 1139 | 1474 | 1107 | 1574 | 1368 |
| EvoAgent | 23929 | 14808 | 22883 | 14862 | 21111 | 19519 |
| AgentPO | 1960 | 1121 | 1724 | 1299 | 1509 | 1522 |

# E PSEUDOCODE

In this section, we provide the pseudocode for the AgentPO algorithm to clarify its implementation and workflow, as described in the main text.

---

**Algorithm 1** AgentPO Algorithm

---

1: **Input:** Collaborator policy $\pi_\theta$, fixed Worker policy $\pi_\phi$, problem distribution $\mathcal{D}$.
2: **Input:** Collaboration Topology $T \in \{\text{Hint-Actor, Critic-Actor}\}$.
3: **Initialize:** Collaborator parameters $\theta$.
4: **for** each training step **do**
5:     Sample a problem $(q, y^*)$ from $\mathcal{D}$.
6:     **if** $T$ is Hint-Actor **then**
7:         Define Collaborator context: $c_\theta \leftarrow q$.
8:         Generate hint (collaborative signal): $z \sim \pi_{\theta_{\text{old}}}(\cdot \mid c_\theta)$.
9:         Worker generates final solution: $y_{\text{final}} \sim \pi_\phi(\cdot \mid [q; z])$.
10:     **else if** $T$ is Critic-Actor **then**
11:         Worker generates initial solution: $y_{\text{init}} \sim \pi_\phi(\cdot \mid q)$.
12:         Define Collaborator context: $c_\theta \leftarrow [q; y_{\text{init}}]$.
13:         Generate critique (collaborative signal): $z \sim \pi_{\theta_{\text{old}}}(\cdot \mid c_\theta)$.
14:         Worker generates final solution: $y_{\text{final}} \sim \pi_\phi(\cdot \mid [q; y_{\text{init}}; z])$.
15:     **end if**
16:     Compute reward: $R \leftarrow \mathbb{I}(y_{\text{final}} = y^*)$.
17:     Store trajectory $(c_\theta, z, R)$ in an experience buffer $\mathcal{B}$.
18:     Update policy $\pi_\theta$ using the GRPO objective on data from $\mathcal{B}$.
19: **end for**
20: **Return:** Optimized Collaborator $\pi_\theta$.

---

# F   LLM USAGE STATEMENT

LLMs were used in the preparation of this paper solely for grammatical correction and language polishing, and they help improve the clarity and fluency of the writing. All intellectual contributions, including concept development, algorithm design, experimentation, and interpretation of results, are entirely the work of the authors.

# G CASE STUDY

## G.1 HINT-ACTOR TOPOLOGY

**Question**: What power of 4 is equal to 8? Express your answer as a common fraction. (Correct answer is $\boxed{\dfrac{3}{2}}$).

**Hint Agent**:
What power of 4 is equal to 8? Express your answer as a common fraction. This question can be simplified to: What fraction $x$ satisfies $4^x = 8$?

**Actor Agent**:
To find the power of 4 that is equal to 8, we can start by expressing both 4 and 8 as powers of 2, since 4 is $2^2$ and 8 is $2^3$. We have: $4 = 2^2$ and $8 = 2^3$. Now, we can rewrite the equation $4^x = 8$ as $(2^2)^x = 2^3$. Using the property of exponents that $(a^b)^c = a^{bc}$, we can simplify the left-hand side of the equation to $2^{2x}$. So, we have: $2^{2x} = 2^3$. Since the bases are the same (2 in this case), we can equate the exponents: $2x = 3$ Now, we can solve for $x$ by dividing both sides by 2: $x = 3/2$. Therefore, the power of 4 that is equal to 8 is 3/2.

The final answer is $\boxed{\dfrac{3}{2}}$.

## G.2 CRITIC-ACTOR TOPOLOGY

**Question**:
The graph of $f(x) = \frac{2x}{x^2-5x-14}$ has vertical asymptotes $x = a$ and $x = b$, and horizontal asymptote $y = c$. Find $a + b + c$.
(Correct answer is $\boxed{5}$).

**Actor Agent (Initial Solution)**:
Step 1: The function $f(x) = \frac{2x}{x^2-5x-14}$ has vertical asymptotes at the values of $x$ that make the denominator equal to zero. To find these values, we need to solve the equation $x^2 - 5x - 14 = 0$. Step 2: The solutions to the equation $x^2 - 5x - 14 = 0$ are $x = -2$ and $x = 7$. These are the values of $a$ and $b$. Step 3: To find the horizontal asymptote, we need to compare the degrees of the numerator and denominator. The degree of the numerator is 1, and the degree of the denominator is 2. Since the degree of the denominator is higher, the horizontal asymptote is $y = 0$. Step 4: Now, we can calculate the sum of a, b, and c. a = -2, b = 7, c = 0.
The final answer is: $\boxed{-9}$.

**Critic Agent**:
1. Logical Gap: The provided solution does not explicitly state that the horizontal asymptote is $y = 0$ based on the degrees of the numerator and denominator.
2. Calculation Error: The sum $a + b + c$ was incorrectly calculated as -9 instead of 5.

**Actor Agent (Final Solution)**:
Step 1: Find the values of a and b. The function $f(x) = \frac{2x}{x^2-5x-14}$ has vertical asymptotes at the values of $x$ that make the denominator equal to zero. To find these values, we need to solve the equation $x^2 - 5x - 14 = 0$. Step 2: Determine the values of a and b.The solutions to the equation $x^2 - 5x - 14 = 0$ are $x = -2$ and $x = 7$. These are the values of $a$ and $b$.
Step 3: Find the value of c. To find the horizontal asymptote, we need to compare the degrees of the numerator and denominator. The degree of the numerator is 1, and the degree of the denominator is 2. Since the degree of the denominator is higher, the horizontal asymptote is $y = 0$. So, $c = 0$.
Step 4: Calculate the sum a + b + c. The sum $a + b + c$ should be: $a + b + c = -2 + 7 + 0 = 5$
The final answer is: $\boxed{5}$.

