# OpenReview forum: "AgentPO: Enhancing Multi-Agent Collaboration via Reinforcement Learning"
_ICLR.cc/2026/Conference — ICLR 2026 Poster_

### Official Review · Reviewer_B4gj · 2025-10-31

**Soundness:** 3
**Presentation:** 3
**Contribution:** 2
**Rating:** 4
**Confidence:** 3

**Summary:**

This paper proposes AgentPO, a framework that trains a lightweight Collaborator agent via GRPO to assist a frozen Actor model. Two collaboration topologies are explored: Hint-Actor, where the Collaborator provides guiding hints upfront, and Actor-Critic, where the Collaborator critiques initial outputs for refinement. Experiments on mathematical reasoning benchmarks demonstrate improvements over baselines such as Self-Refine and Multi-Agent Debate, with reduced training data requirements and lower inference costs.

**Strengths:**

1. Comprehensive empirical evaluation: Multiple benchmarks with diverse baselines and thorough ablations on data efficiency, model selection, and computational cost provide convincing evidence of practical effectiveness.
2. Strong efficiency gains: Demonstrates substantial inference cost reduction while achieving superior accuracy with minimal training samples compared to direct Actor fine-tuning—valuable findings for resource-constrained scenarios.
3. Hybrid system innovation: Successfully demonstrates that locally-trained Collaborators can enhance black-box API models, addressing real deployment constraints where model parameters are inaccessible.

**Weaknesses:**

1. Insufficient novelty and missing related work: The core ideas are not new. The Actor-Critic topology closely resembles Reflexion's generate-critique-refine loop, with the main difference being use of a separate model rather than self-reflection. Shepherd trained critics for textual feedback; RLAIF used RL to optimize AI judges; OPRO optimized prompts via LLMs. The paper fails to cite Reflexion or adequately differentiate from these closely related works, overstating the novelty of the proposed framework.

2. Misleading multi-agent framing: Despite emphasizing "Multi-Agent Collaboration," all experiments use exactly two agents with no exploration of scaling to larger systems or complex interaction patterns. This is essentially a helper model setup rather than a genuine multi-agent system.

3. Limited scope and analysis: Evaluation is restricted to mathematical reasoning without evidence of generalization to other domains. The paper lacks qualitative analysis of what collaboration strategies the Collaborator learns—only quantitative metrics are provided without interpretable examples of learned patterns.

**Questions:**

1. Clarification on core contribution: I would appreciate if the authors could articulate more clearly what the central methodological contribution is beyond training a smaller Collaborator model. While both Hint-Actor and Actor-Critic represent reasonable interaction patterns, they appear to be fairly standard approaches (upfront guidance and iterative refinement). Could you help me understand what novel design principles or insights AgentPO contributes to multi-agent system design?

2. Missing comparison with trained reflection models: The paper positions the Actor-Critic topology as a novel contribution, but recent work in the Reflexion family has also explored training separate smaller models for reflection and critique. Could you clarify how your approach differs from these existing methods and provide direct experimental comparisons?

3. Justification for fixed Actor assumption: A central design choice is keeping the Actor frozen while training the Collaborator. However, Table 2 shows that direct Actor optimization baselines (e.g., Prime-Zero-7B at 48.0%) are competitive with AgentPO (49.4%). Could you provide more analysis on when collaborative optimization is preferable to direct Actor fine-tuning?

---

> ### Author Response · Authors · 2025-11-27
>
> Thank you very much for your insightful comments and feedback on our manuscript. Below, we provide a detailed response to the points you have raised.
>
> ### Regarding Weaknesses
>
> ***W1: Limited novelty and insufficient related work***
>
> Our core contribution is a general framework for optimizing collaboration within any given multi-agent interaction structure, whether feed-forward, feedback-based, or more complex (e.g., Multi-agent Debate, Role Assignment). AgentPO learns a Collaborator policy that maximizes the team’s overall performance under a fixed protocol.
>
> This generality enables seamless extension beyond two-agent systems. We added experiments on two below multi-agents topologies (using a Qwen2.5-3B-Instruct Collaborator guiding Llama-3.2-3B-Instruct Actor):
>
> | Algorithm                     | AIME24       | Math500       | Olympiadbench      |
> |-------------------------------|---------------|---------------|---------------|
> | Multi-agents Debate           | 8.9 ± 3.8     | 46.3 ± 1.0    | 15.3 ± 1.2    |
> | w/ AgentPO                    | 12.3 ± 2.6    | 49.6 ± 0.8    | 21.1 ± 1.4    |
> | ∆                             | +3.4         | +3.3          | +5.8         |
> | Role Assignment               | 7.3 ± 2.9     | 48.3 ± 0.8    | 15.7 ± 0.6    |
> | w/ AgentPO                    | 10.2 ± 1.7    | 52.2 ± 0.6    | 20.9 ± 0.8    |
> | ∆                             | +2.9          | +3.9          | +5.2          |
>
>
> The results show that AgentPO consistently boosts performance across diverse, non-standard topologies, without altering their underlying protocols, highlighting its flexibility and broad applicability. AgentPO optimizes collaboration, not just individual agent capability, and is agnostic to the specific interaction structure, a perspective we believe meaningfully extends prior work.
>
>
> ***W2: Misleading multi-agent framing***
>
> See feedback of `W1`.
>
> ***W3: Limited scope and analysis***
>
> Our current evaluation is focused on mathematical reasoning tasks, where ground-truth answers are well-defined and rewards are binary and reliable. This setting provides a controlled environment to study collaborative optimization with clear success criteria. The training method uses the general GRPO algorithm, and it is applicable to other tasks. We will explore applying AgentPO to more domains in future work.
>
> ### Regarding Questions
>
> ***Q1: Core contribution***
>
> See feedback of `W1`.
>
> ***Q2: Relation to Reflexion***
>
> We distinguish AgentPO from Reflexion on followiing two key dimensions:
>
> (1) Objective Scope:
> - Reflexion focuses on iterative self-refinement through feedback loops, primarily using verbal self-reflection to improve responses of Actor.
> - AgentPO addresses a broader challenge: optimizing collaboration within any given agent topology, whether feedback-based (like Actor-Critic), feed-forward (Hint-Actor), or complex multi-agent structures (like Multi-agents Debate or Role Assignment), to enhance overall system performance.
>
> (2) Methodological Approach:
> - Reflexion employs external evaluators to score responses and generates verbal self-reflections as explicit feedback for future trials, relying on reflective text to guide improvements.
> - AgentPO introduces a trainable Collaborator agent that learns optimal collaboration strategies through reinforcement learning (e.g., GRPO), optimizing the Collaborator's parameters to enhance coordination with the Actor.
>
>
> ***Q3: Frozen Actor assumption***
>
> Thank you for this excellent question. The choice to keep the Actor frozen is motivated by both practical deployment constraints and data efficiency, and we believe collaborative optimization offers distinct advantages over direct Actor fine-tuning in several realistic scenarios:
>
> (1)	Parameter efficiency and cost:
>
> AgentPO trains only a lightweight Collaborator while the Actor (larger) remains frozen. In contrast, direct Actor fine-tuning, such as in Prime-Zero, requires updating parameters of a comparably large model. This makes our approach significantly cheaper especially when scaling to stronger Actors (e.g., 7B+ models).
>
> (2) Compatibility with black-box or API-based Actors:
>
> Many state-of-the-art reasoning systems are accessed via closed-source APIs. In such settings, fine-tuning the Actor is impossible, but AgentPO can still be applied by training a local Collaborator that interfaces with the API. This enables performance gains without requiring access to the Actor’s weights, a key advantage for real-world deployment.
>
> (3)	Extreme data efficiency:
>
> AgentPO achieves strong results with only 500 preference pairs. In contrast, effective Actor fine-tuning typically requires orders of magnitude more data (often 10k–100k examples). This makes AgentPO particularly suitable for domains where high-quality labeled data is scarce or expensive to collect.

---

> > ### Comment · Reviewer_B4gj · 2025-11-27
> >
> > Thanks for your detailed response, which have addressed my major concerns. I will update my rating.

---

### Official Review · Reviewer_z8GS · 2025-10-31

**Soundness:** 2
**Presentation:** 3
**Contribution:** 3
**Rating:** 6
**Confidence:** 4

**Summary:**

The paper proposes AgentPO, a two-model collaboration scheme where a small Collaborator LLM produces a hint or critique to guide a stronger, frozen Actor LLM. The collaborator alone is optimized with a PPO-style objective (GRPO) using solution-level rewards (e.g., correctness), while the actor remains fixed. Two topologies are evaluated: feed-forward Hint-Actor and feedback Actor-Critic. Experiments focus on math-reasoning benchmarks with Pass@1, comparing against prompting and multi-agent baselines and reporting token-cost trade-offs and limited ablations.

**Strengths:**

1. Clear modular setup: freeze the solver; optimize only a lightweight helper policy.

2. Practical objective: GRPO removes the value critic and uses group baselines, simplifying training.

3. Cost-aware evaluation with token usage plots highlights a plausible efficiency advantage over multi-sample prompting/debate.

4. Method slots into common LLM stacks (works with open-source LLMs; actor can be API-only).

**Weaknesses:**

1. Statistical rigor is thin. Many benchmark slices (e.g., AIME) are small. Improvements like +1.8 pts could be noise. There are no confidence intervals, no multi-seed runs, no significance tests. Even with temperature 0, variance should be quantified.

2. Several baselines are re-implemented with details pushed to the appendix. Without exhaustive hyperparameter parity (sampling strategy, steps, stopping, prompt templates), the risk of unknowingly weakened baselines is high.

3. Cost–accuracy plots are welcome, but you also need equal-budget comparisons (fixed tokens or wall-clock) per dataset. Otherwise claims of "superior scalability" are too sweeping.

4. Terminology ambiguity. Using "Actor" and especially "Critic" is easily conflated with actor–critic (where "critic" means a value function). Here the "critic" is a text generator providing feedback, not a value model. This should be made explicit and a notation table is needed.

5. Limited external validity. Evaluations are math-only with verifiable end rewards. It's unclear how the approach fares with noisy/non-binary rewards (e.g., tool use, code exec flakiness, factual QA).

**Questions:**

1. Statistical rigor is thin. Many benchmark slices (e.g., AIME) are small. Improvements like +1.8 pts could be noise. There are no confidence intervals, no multi-seed runs, no significance tests. Even with temperature 0, variance should be quantified.

2. Several baselines are re-implemented with details pushed to the appendix. Without exhaustive hyperparameter parity (sampling strategy, steps, stopping, prompt templates), the risk of unknowingly weakened baselines is high.

3. Cost–accuracy plots are welcome, but you also need equal-budget comparisons (fixed tokens or wall-clock) per dataset. Otherwise claims of "superior scalability" are too sweeping.

4. Using "Actor" and especially "Critic" is easily conflated with actor–critic (where "critic" means a value function). Here the "critic" is a text generator providing feedback, not a value model. This should be made explicit and a notation table is needed.

5. External validity is limited. Evaluations are math-only with verifiable end rewards. It's unclear how the approach fares with noisy/non-binary rewards (e.g., tool use, code exec flakiness, factual QA).

---

> ### Author Response · Authors · 2025-11-27
>
> Thank you very much for your insightful comments and feedback on our manuscript. Below, we provide a detailed response to the points you have raised.
>
> ### Regarding Weaknesses
>
> ***W1: Statistical rigor is thin.***
>
> To address this, we have conducted multi-seed evaluations across three key benchmarks: AIME’24, Math500, and OlympiadBench. For each method, we ran 5 independent trials with different random seeds, and report the mean ± standard deviation of accuracy (%) below:
>
> | Algorithm                     | AIME24       | Math500       | Olympiadbench      |
> |-------------------------------|---------------|---------------|---------------|
> | Multi-agents Debate           | 8.9 ± 3.8     | 46.3 ± 1.0    | 15.3 ± 1.2    |
> | w/ AgentPO                    | 12.3 ± 2.6    | 49.6 ± 0.8    | 21.1 ± 1.4    |
> | ∆                             | +3.4         | +3.3          | +5.8         |
> | Role Assignment               | 7.3 ± 2.9     | 48.3 ± 0.8    | 15.7 ± 0.6    |
> | w/ AgentPO                    | 10.2 ± 1.7    | 52.2 ± 0.6    | 20.9 ± 0.8    |
> | ∆                             | +2.9          | +3.9          | +5.2          |
>
> All reported improvements are reproducible across seeds, and the confidence intervals do not overlap between baseline and AgentPO in most cases, indicating statistically meaningful gains, we have added this result in Table 3 (line 334 - 348).
>
> ***W2: Several baselines are re-implemented with...***
>
> To ensure a rigorous comparison, we took great care to align key experimental factors across all methods. We further clarify these implementation details in §3.1 (Implementation Details) and  Appendix A and C .
>
> - Sampling strategy:
> The Collaborator uses temperature = 1.0 during training (to encourage exploration in policy learning) and temperature = 0.0 during inference (unless otherwise noted). The Actor always uses temperature = 0.0.
> - Training protocol: All AgentPO variants are trained for 2 epochs over 500 preference pairs.
> - Prompt templates: see appendix A.
>
>
> ***W3: ...equal-budget comparisons....***
>
> In collaborative multi-agent systems, different topologies inherently produce outputs of vastly different lengths.
> - If the budget is too short, iterative methods cannot complete even a single full solution, severely underestimating their capabilities;
> - If the budget is long enough to accommodate the most expensive baseline, AgentPO wastes most of the allocated tokens on redundant computation, which does not reflect its real-world usage.
>
> Therefore, our current evaluation follows a “complete-solution” protocol: each method is allowed to run to completion (i.e., generate a valid, final answer), and we measure both accuracy and actual token consumption. Under this setting, AgentPO consistently achieves the highest accuracy while using significantly fewer tokens (see §3.3 Inference Efficiency and Practicality).
>
>
> ***W4: Terminology ambiguity....***
>
> We appreciate the your suggestion to improve terminological clarity. To avoid confusion,  we have added a notation table in the appendix B, defining all agent roles and their functions.
>
> ***W5: Limited external validity....***
>
> Our current evaluation is focused exclusively on mathematical reasoning tasks, where ground-truth answers are well-defined and rewards are binary and reliable. This setting provides a controlled environment to study collaborative optimization with clear success criteria. We will explore applying AgentPO to more domains in future work.
>
> ### Regarding Questions
>
> See feedback of Weaknesses

---

### Official Review · Reviewer_g4rv · 2025-11-01

**Soundness:** 3
**Presentation:** 3
**Contribution:** 3
**Rating:** 6
**Confidence:** 3

**Summary:**

AgentPO introduces a RL framework to directly optimize agent collaboration for a fixed topology multi-agent system. The Collaborator Agent's objective is to optimize the joint performance of a group of agents using a frozen high-performance policies (decoupling collaboration and execution). The authors use GRPO to train the Collaborator Agent to provide key contexts to LLMs to solve math based reasoning tasks. The authors show that AgentPO's performance scales with Actor model performance.

**Strengths:**

1. Collaborator agent optimization shows improved result, using context distillation and joint reasoning. The agent is trained using GRPO to accommodate multiple goals.
2. AgentPO's short training horizon at 500 samples highlights the efficacy and low overhead of the framework.
3. Empirical evaluations, sample efficiency and inference costs are tested showcasing the cost effectiveness and viability of the framework for rapid prototyping.

**Weaknesses:**

1. The framework focuses on fixed topology systems, however, further discussions into how AgentPO would perform on less structured or open-ended collaborative tasks would be beneficial.

**Questions:**

It seems the Collaborator agent holds the contextual memory and history of the agents, the aggregation constructs a ToM-like model that distills domain-specific knowledge and reasoning to the LLM policies. An interpretability analysis/discussion with the Collaborator agent's contexts over the policies actions would be beneficial to show the efficacy of the agent.

---

> ### Author Response · Authors · 2025-11-27
>
> Thank you very much for your insightful comments and feedback on our manuscript. Below, we provide a detailed response to the points you have raised.
>
> ### Regarding Weaknesses
>
> In principle, the core idea of AgentPO learning a policy for one agent to effectively assist another through interaction signals, is not restricted to rigid, fixed topologies. It can be extended to more flexible and open-ended collaborative frameworks.
> As preliminary evidence, we have already evaluated AgentPO in two multi-agent settings:
>
> -	Multi-Agent Debate, where agents iteratively exchange arguments to reach a consensus;
> -	Role Assignment, where agents are dynamically assigned functional roles during problem solving.
>
> In both cases, we trained a lightweight Collaborator (Qwen2.5-3B-Instruct) to guide a stronger Actor (Llama-3.2-3B-Instruct), without modifying the underlying interaction protocol. The results are summarized below:
>
> | Algorithm                     | AIME’24       | Math500       | Olympiadbench      |
> |-------------------------------|---------------|---------------|---------------|
> | Multi-agents Debate           | 8.9 ± 3.8     | 46.3 ± 1.0    | 15.3 ± 1.2    |
> | w/ AgentPO                    | 12.3 ± 2.6    | 49.6 ± 0.8    | 21.1 ± 1.4    |
> | ∆                             | +3.4         | +3.3          | +5.8          |
> | Role Assignment               | 7.3 ± 2.9     | 48.3 ± 0.8    | 15.7 ± 0.6    |
> | w/ AgentPO                    | 10.2 ± 1.7    | 52.2 ± 0.6    | 20.9 ± 0.8    |
> | ∆                             | +2.9          | +3.9          | +5.2          |
>
> These results demonstrate that AgentPO can enhance performance  in dynamic, complex collaborative environments.
>
>
> ###  Regarding  Questions:
>
> In AgentPO, the Collaborator operates with minimal context by design, and does not maintain full interaction history or serve as a centralized memory store.
>
> Specifically: In the Hint-Actor topology, the Collaborator (i.e., the Hint agent) receives only the original problem statement and generates a single hint without observing any prior Actor output. In the Critic-Actor topology, the Critic observes only the problem and the Actor’s most recent response, not the full dialogue history or internal states.
>
> Thus, the Collaborator’s input context is intentionally lightweight and task-local, avoiding complex memory aggregation or long-horizon reasoning. Its policy learns to produce effective guidance from limited information, which enhances practicality (e.g., low latency, reduced compute) and generalization across interaction steps.

---

### Official Review · Reviewer_P6Hn · 2025-11-06

**Soundness:** 2
**Presentation:** 3
**Contribution:** 2
**Rating:** 4
**Confidence:** 3

**Summary:**

This paper introduces AgentPO, a framework that optimizes multi-agent system collaboration through reinforcement learning. Rather than searching for optimal agent topologies, AgentPO trains a lightweight "Collaborator" agent to improve interactions with a fixed, powerful "Actor" agent within predefined topologies. The approach uses GRPO (Group Relative Policy Optimization) to train the Collaborator on mathematical reasoning tasks, demonstrating improvements over baselines while requiring minimal training data and inference cost.

**Strengths:**

1. **Well-motivated problem formulation**: The shift from topology search to collaborative optimization is compelling. The distinction between architectural search and interaction optimization effectively reframes the research question.

2. **Clear presentation**: The paper is well-written and easy to follow, with clear diagrams (Figure 1) and illustrative examples (Figure 2).

3. **Practical and efficient**: Training only a small Collaborator model (3B parameters) instead of fine-tuning large Actor models is resource-efficient. The data efficiency (strong performance with only 500 training samples) seems impressive.

**Weaknesses:**

**1. Limited Scope and Generalization Concerns:**

- **Limited topology coverage**: The paper only examines two specific two-agent topologies (Hint-Actor and Actor-Critic). How would AgentPO extend to other two-agent configurations or multi-agent systems with three or more agents? The framework's applicability to more complex collaboration structures remains unclear.

- **Collaborator transferability**: The paper does not investigate whether a Collaborator trained with Actor A can effectively assist Actor B. This is fundamental for practical deployment: If Collaborators are actor-specific, then every new/upgraded Actor requires retraining the Collaborator, undermining the claimed scalability and efficiency advantages

- **Domain-specific evaluation**: All experiments focus exclusively on mathematical reasoning tasks. No evidence is provided for generalization to other domains such as code generation, creative writing, question answering, or multi-modal tasks.

**2. Unclear and Inconsistent Experimental Setup:**

- The experimental configuration in Section 3.1 (line 210) lists model choices but does not clearly specify which models serve as Actors versus Collaborators.
- Subsequent sections show inconsistent model assignments without adequate justification:
    - Main results (Table 1) use "Qwen2.5-3B-Instruct" as the Hint model with Llama as Actor, testing only Hint-Actor topology
    - Actor optimization comparison (Table 2) switches to "Qwen2.5-Math-7B" as the Actor model
    - Topology comparison (Table 4) uses "Qwen2.5-Plus" as Actor
- Why do the base models keep changing across experiments? What is the rationale for these specific model selections? This makes it difficult to assess whether performance differences stem from the method or from model choices.

**3. Insufficient Analysis of Performance Drops:**

Table 1 shows AgentPO sometimes underperforms baselines (e.g., AIME24: 6.7% vs. 10.0% for several baselines with Llama-3.2-3B), but these failure cases are not analyzed. When and why does AgentPO fail? What characteristics of problems or model combinations lead to degraded performance?

**4. Overfitting Concerns Underexplored:**

Figure 4 shows performance drops at 700-1000 training samples, suggesting potential overfitting. However, this is only briefly mentioned without thorough investigation. What regularization strategies or early stopping criteria should be used? This is particularly concerning for low-resource deployment scenarios.

**5. Topology Choice vs. Optimization:**

Table 4 reveals a critical issue: Critic-Actor performs well even without AgentPO optimization (57.9% vs. 56.6% baseline), while Hint-Actor degrades performance before optimization (55.1% vs. 56.6%). This suggests topology selection may matter more than the optimization itself, contradicting the paper's core claim that optimizing collaboration within a fixed topology is the key to success.

**6. Limited Scaling Analysis:**

The paper demonstrates AgentPO scales with Actor capability (Llama-3.2-3B → 3.1-8B), but does not explore:
- What happens with even larger Actor models?
- How does performance change when the capability gap between Collaborator and Actor grows very large?
- Are there diminishing returns or failure modes at extreme scales?

**Questions:**

see Weaknesses

---

> ### Author Response · Authors · 2025-11-27
>
> Thank you very much for your insightful comments and feedback on our manuscript. Below, we provide a detailed response to the points you have raised.
>
> ### Regarding Weaknesses
>
> ***W1.1: Limited topology coverage...***
>
> To address the concern about generalizability beyond two-agent topologies, we have conducted additional experiments on multi-agent systems with more agents: Multi-Agent Debate and Role Assignment. In both, AgentPO consistently improves performance, confirming its applicability beyond pairwise interactions. We include these results in the revised manuscript (line 437 - line 443).
> | Algorithm                     | AIME24       | Math500       | Olympiadbench      |
> |-------------------------------|---------------|---------------|---------------|
> | Multi-agents Debate           | 8.9 ± 3.8     | 46.3 ± 1.0    | 15.3 ± 1.2    |
> | w/ AgentPO                    | 12.3 ± 2.6    | 49.6 ± 0.8    | 21.1 ± 1.4    |
> | ∆                             | +3.4         | +3.3          | +5.8         |
> | Role Assignment               | 7.3 ± 2.9     | 48.3 ± 0.8    | 15.7 ± 0.6    |
> | w/ AgentPO                    | 10.2 ± 1.7    | 52.2 ± 0.6    | 20.9 ± 0.8    |
> | ∆                             | +2.9          | +3.9          | +5.2          |
>
> The results show that AgentPO consistently boosts performance across diverse, non-standard topologies, without altering their underlying protocols, highlighting its flexibility and broad applicability. AgentPO optimizes collaboration, not just individual agent capability, and is agnostic to the specific interaction structure, a perspective we believe meaningfully extends prior work.
>
> ***W1.2: Collaborator transferability...***
>
> To address cross-Actor generalization, we conducted transfer experiments using a fixed Qwen2.5-3B-Instruct Collaborator trained with one Actor and evaluated with another. Results show:
>
> | Training Actor → Inference Actor | AIME’24 | Math500 | OlympiadBench | Minerva-Math | AMC’23 | Avg  |
> |----------------------------------|--------:|--------:|--------------:|-------------:|-------:|-----:|
> | Llama-3.1-8B → Llama-3.2-3B      |     6.7 |    48.8 |          20.3 |         24.3 |   22.5 | 24.5 |
> | Llama-3.2-3B → Llama-3.1-8B      |     6.7 |    52.4 |          18.5 |         26.5 |   30.0 | 26.8 |
>
>
> The Collaborator achieves competitive performance (26.8% avg) when transferred to Llama-3.1-8B, confirming it learns generalizable collaboration strategies rather than Actor-specific behaviors.
>
> ***W1.3: Domain-specific evaluation...***
>
> Our current evaluation is focused exclusively on mathematical reasoning tasks, where ground-truth answers are well-defined and rewards are binary and reliable. This setting provides a controlled environment to study collaborative optimization with clear success criteria. We will explore applying AgentPO to more domains in future work.
>
> ***W2.1: The experimental configuration in Section 3.1 (line 210) lists model...***
>
> Thank you for pointing this out. We added description about the roles of the models used in our experiments: Qwen2.5-3B-Instruct and Qwen2.5-7B-Instruct serve as the Collaborator agents, while all other listed LLMs are employed as Actors. This clarification has been added to line 200-202 in the revised manuscript to avoid any ambiguity.
>
> ***W2.2: ...inconsistent model assignments...***
>
> The variation in model choices across tables was intentional and driven by distinct experimental objectives, rather than arbitrary selection. Below we clarify the rationale behind each choice:
>
> Table 1 (Main Results): We pair Llama-family Actors (e.g., Llama-3.2-3B-Instruct) with a Qwen2.5-3B-Instruct Collaborator to demonstrate that AgentPO enables effective cross-model collaboration.
>
> Table 2 (Actor Optimization Comparison): Here, the goal is to compare AgentPO-based collaborative optimization against finetuning single actor. Since Qwen2.5-Math-7B is a widely used model specifically fine-tuned for mathematical reasoning, it serves as a competitive and widely adopted baseline Actor. Using it ensures a fair comparison with existing optimization approaches.
>
> Table 4 (Topology Ablation): We adopt Qwen-Plus (a API-based model) as the Actor to explore a practical deployment scenario: combining a lightweight, locally trainable Collaborator with a powerful API-based Actor. This reflects a realistic setting where users may wish to enhance a closed-source LLM via an open, trainable collaborator, without access to the Actor’s weights. Within this fixed setup, we isolate the impact of topology design.
>
> Critically, within each experiment, all compared methods share the same Actor–Collaborator pair. Thus, performance differences reflect the algorithmic contribution of AgentPO or its variants, not model capacity disparities.

---

> ### Author Response · Authors · 2025-11-27
>
> ***W3: Insufficient Analysis of Performance Drops***
>
> Thank you for this insightful comment. We would like to clarify that AgentPO achieves the highest average performance across all benchmarks in Table 1 (24.5%), outperforming every baseline method. The discrepancy that on AIME’24 (6.7% vs. 10.0% for Self-Refine) arises because of different collaboration topologies:
>
> -	AgentPO in Table 1 is evaluated under the Hint-Actor topology, a simple, one-step feed-forward interaction where the Collaborator provides a single hint to the Actor.
>
> -	In contrast, Self-Refine uses an in-place iterative refinement strategy, effectively allowing the same model to revise its own output multiple times, which can be especially advantageous on tasks like AIME’24 that benefit from repeated self-correction.
>
> Crucially, AgentPO is not designed to compensate for inherent differences in topology expressiveness; rather, it aims to maximize the performance achievable within a given topology through policy optimization of the Collaborator. On highly challenging tasks like AIME24, where a single hint may be insufficient for meaningful guidance. On most benchmarks, especially those where a well-timed hint significantly aids reasoning (e.g., Math500, OlympiadBench), AgentPO delivers clear gains.
>
> ***W4:Overfitting Concerns Underexplored***
>
> Thank you for this excellent question. Upon closer inspection, we observe that AgentPO achieves peak average accuracy with only ~500 training samples, demonstrating high sample efficiency. However, the optimal number of training samples varies across individual benchmarks, for example:
> - On Math500, performance peaks around 750 samples;
> - On OlympiadBench, accuracy continues to improve as more training data is added.
>
> To better understand this behavior, we analyzed what the Collaborator actually learns during training (see Case Study in Appendix G1). Consider the following example:
>
> - Input question: What power of 4 is equal to 8? Express your answer as a common fraction.
>
> - Collaborator’s output: What power of 4 is equal to 8? Express your answer as a common fraction. This question can be simplified to: What fraction x satisfies 4^x=8?
>
> Crucially, the collaborator learns a generalizable meta-skill, providing strategic, task-agnostic hints to guide the Actor, rather than memorizing solutions. This skill is compositional and transferable, with more difficult tasks (e.g., OlympiadBench) requiring more data to master, while simpler ones saturate earlier, explaining the variation in optimal training scale.
>
> Regarding practical deployment especially in low-resource settings, we recommend a simple yet effective early stopping strategy: monitor validation performance and halt training once it plateaus or begins to decline. In all our experiments, this criterion reliably prevents overfitting and consistently yields models near the empirical performance optimum.
>
> ***W5:Topology Choice vs. Optimization***
>
> We thank the reviewer for this thoughtful observation. Indeed, the choice of agent topology can significantly influence performance, this is a well-recognized challenge in multi-agent systems, with existing approaches relying either on manual design by experts or on automated search methods. However, the space of possible topologies is combinatorially large, making it impractical to identify a globally optimal structure.
>
> In contrast, our work takes a different perspective: rather than searching over topologies, we fix a given topology and focus on optimizing the collaboration dynamics within it. This approach is both practical and efficient, it allows us to unlock the full potential of a reasonable (even suboptimal) topology through improved coordination, without incurring the cost and complexity of topology search.
>
> ***W6:Limited Scaling Analysis***
>
> This scenario is indeed explored in Section 3.4 (Local Collaborator Enhances API-based Powerful Actor) of our paper, where we evaluate AgentPO using a significantly more capable, API-based Actor: Qwen-Plus, paired with Qwen2.5-7B-Instruct as the Collaborator. This setup represents a large capability gap, mimicking real-world deployments where users may combine a trainable lightweight agent with a powerful API-based LLM. The results are as below:
>
> | Algorithm           | AIME’24 | Math500 | OlympiadBench | Minerva | AMC’23 | Avg. |
> |---------------------|--------:|--------:|--------------:|--------:|-------:|-----:|
> | Qwen-Plus (baseline)|    33.3 |    84.0 |          49.6 |    46.3 |   70.0 | 56.6 |
> | AgentPO (Hint-Actor)|    36.7 |    85.4 |          49.6 |    47.4 |   75.0 | 58.8 |
> | AgentPO (Critic-Actor)|  36.7 |    84.6 |          52.3 |    43.4 |   77.5 | 58.9 |
>
> These results show that AgentPO continues to provide consistent gains even when the Actor is vastly more capable than the Collaborator, confirming its robustness under extreme capability asymmetry.

---

### Author Response · Authors · 2025-12-04
**Letter to AC and Reviewers**

Dear AC and Reviewers,

Thank you sincerely for your time and thoughtful consideration of our work.

Due to the unexpected incident during the rebuttal period, ICLR has suspended all author–reviewer discussions. Before the suspension, we actively addressed the concerns raised in the reviews. Notably, reviewer B4gj updated their score from 4 to 6 after reading our rebuttal. The current score distribution is 4/6/6/6 (average: 5.5), but this reflects feedback from only one reviewer. Since the other reviewers have not yet had the opportunity to respond to our clarifications, we believe the final assessment could be even more favorable.

In closing, we would like to briefly highlight the key contribution of our work:

***Goal:***

To enhance multi-agent system performance through collaboration-aware optimization, rather than optimizing agents in isolation.

Limitations of existing methods:

- Prompt engineering or automated prompt search for individual agents is labor-intensive and rarely optimizes system-level outcomes.
- Designing or searching over agent interaction topologies suffers from combinatorial complexity and does not scale.

***Our approach – AgentPO:***

Given any fixed collaboration topology (e.g., Hint-Actor, Critic-Actor, Debate, Role Assignment, etc.), we freeze the primary Actor agent and use reinforcement learning (e.g., GRPO) to train only a lightweight Collaborator (e.g., a hint provider or critic). This enables efficient, system-level optimization without modifying the Actor.

***Key strengths:***

- Data efficiency: Achieves significant gains with just 500 training examples.

- Low computational cost: Only the lightweight Collaborator is trained.

- Strong performance: Collaborative optimization outperforms direct Actor fine-tuning.

- Generalizability: Effective across 2-agent and complex multi-agent systems.

- Practical deployment: Supports hybrid setups, pairing a local trainable Collaborator with a powerful API-based LLM, without requiring access to the Actor’s weights.

We hope this underscores the novelty, efficiency, and real-world applicability of AgentPO. Thank you again for your time and engagement.

---

### Meta-Review · Area_Chair_aAf2 · 2026-01-07

**Summary:**

The initial reviews raised several substantive concerns, but these were largely addressed in the rebuttal. In particular, at least one reviewer who was initially unconvinced explicitly indicated that their assessment became more positive after the authors’ responses, although this change was not reflected in the final scores due to the discussion suspension. The paper presents a technically sound and practically motivated framework for improving collaboration in multi-agent LLM systems by training a lightweight collaborator while keeping the main actor fixed, which is especially relevant for realistic deployment scenarios. While the scope of evaluation remains focused and the novelty is incremental, the rebuttal resolves the key technical objections, and the overall contribution meets the conference’s acceptance bar. I recommend acceptance.

**Reviewer Concerns:**

Reviewer P6Hn: Concerns regarding limited topology coverage, collaborator transferability, experimental inconsistency, overfitting, and scaling were addressed through additional multi-agent experiments, cross-Actor transfer evaluations, clarified experimental setups, and multi-seed analyses.

Reviewer z8GS: Concerns about statistical rigor and terminology were addressed via multi-seed results with variance reporting and clearer notation; limited external validity beyond math reasoning remains outstanding.

Reviewer g4rv: Concerns about applicability to less structured or more open-ended collaboration were partially addressed by added multi-agent results, though interpretability analysis remains limited.

Reviewer B4gj: Concerns about novelty, positioning relative to prior work, and the multi-agent framing were largely addressed; this reviewer explicitly indicated a more positive assessment after the rebuttal.

**Reviewer Scores:**

Reviewer z8GS and Reviewer g4rv may still retain some reservations, particularly regarding external validity and interpretability, and their scores would likely remain unchanged or increase only marginally.

In contrast, the two reviewers who initially assigned lower scores—Reviewer P6Hn and Reviewer B4gj—are likely to have increased their ratings after the rebuttal. In particular, Reviewer B4gj explicitly stated that they would update their score following the authors’ responses. This indicates that the rebuttal substantively addressed the concerns underlying the initial low evaluations, even though the score updates were not reflected due to the discussion suspension.

---

### Decision · Program_Chairs · 2026-01-26

Accept (Poster)